# Development of a New Ecological Magnetic Abrasive Tool for Finishing Bio-Wire Material

**DOI:** 10.3390/ma12050714

**Published:** 2019-03-01

**Authors:** Cheng Yin, Lida Heng, Jeong Su Kim, Min Soo Kim, Sang Don Mun

**Affiliations:** Division of Mechanical Design Engineering, Chonbuk National University, 664-14, Duckjin-gu, Jeonju 561-756, Korea; yoonsung_ys@hotmail.com (C.Y.); henglida1@gmail.com (L.H.); jeongsu1592@naver.com (J.S.K.); kimms@jbnu.ac.kr (M.S.K.)

**Keywords:** wire magnetic abrasive finishing, 316L SUS wire, ecological magnetic abrasive tools, surface roughness

## Abstract

This study proposes a new wire magnetic abrasive finishing (WMAF) process for finishing 316L SUS wire using ecological magnetic abrasive tools. 316L SUS wire is a biomaterial that is generally used in medical applications (e.g., coronary stent, orthodontics, and implantation). In medical applications of this material, a smooth surface is commonly required. Therefore, a new WMAF process using ecological magnetic abrasive tools was developed to improve the surface quality and physical properties of this biomaterial. In this study, the WMAF process of 316L SUS wire is separated into two finishing processes: (i) WMAF with ecological magnetic abrasive tools, and (ii) WMAF with industrial magnetic abrasive tools. The ecological magnetic abrasive tools consist of cuttlefish bone abrasives, olive oil, electrolytic iron powder, and diamond abrasive paste. The finishing characteristics of the two types of abrasive tools were also explored for different input parameters (i.e., vibrating magnetic field and rotating magnetic field). The results show that ecological magnetic abrasive tools can improve the initial surface roughness of 316L SUS wire from 0.23 µm to 0.06 µm. It can be concluded that ecological magnetic abrasive tools can replace industrial magnetic abrasive tools.

## 1. Introduction

316L SUS wire is a biomaterial that is commonly used for medical applications (e.g., coronary stent, orthodontics, and implantation) because of its superior mechanical properties, physical properties, and anticorrosion properties [1,2]. A high-quality surface finish is required in medical applications of this material. However, it is difficult to achieve a high-quality surface on these wire biomaterials by conventional finishing techniques [3,4,5,6]. Previously, surface improvement techniques such as electropolishing (ECP) and ion implantation have been used to improve the surface quality and physical properties of 316L SUS wire material. These methods can achieve a high-quality surface finish on biomaterials. Despite their demonstrated success, there are some inconveniences and limitations during and after electropolishing. Lee et al. [7] reported that electropolishing cannot smear over or otherwise conceal defects such as seams and non-metallic inclusions in metals. In addition, heavy orange peel texture, mold-surface texture and rough scratches are not removed by a practical amount of electropolishing, and thus require an initial "cutdown" with abrasives. Smick et al. [8] reported that ion implantation uses extremely toxic gas sources such as phosphine and arsine, which means that it can have significant effects on human health.

The limitations of these methods are as follows:Rough surface defects cannot be removed.Due to selective dissolution of different phases, electropolishing of multiphase alloys may cause roughening.During the process, they use some chemicals that leave a long-lasting impact on the material finish.

To overcome these inconveniences and limitations, a new wire magnetic abrasive finishing process using ecological magnetic abrasive tools is devised. This process can effectively achieve surface accuracy and dimensional accuracy of a wire workpiece such as 316L SUS wire. Because 316L SUS wire has generally been used in medical applications, the industrial magnetic abrasive tools that are widely used in the finishing process were replaced with ecological magnetic abrasive tools. In the conventional finishing process, when industrial magnetic abrasive tools were used, toxic substances used in the lubricant and abrasive tools were likely to remain on the surface of the material after processing [9]. When finishing biomaterials, a high-quality surface finish and a lack of toxic substances on the surface finish of the material are required.

In this study, wire magnetic abrasive finishing of 316L SUS is separated into two types of finishing processes: (i) WMAF with ecological magnetic abrasive tools and (ii) WMAF with industrial magnetic abrasive tools. The mixture used in the ecological magnetic abrasive tools consists of iron particles, diamond paste, cuttlefish bone abrasive and olive oil. In contrast, the mixture used with the industrial magnetic abrasive tools consists of iron particles, diamond paste, CNT abrasive and light oil.

We explored a new wire magnetic abrasive finishing technique using ecological magnetic abrasive tools to achieve the required surface quality of 316L SUS wire material, and then the finishing performances of these different magnetic abrasive tools (i.e., ecological magnetic abrasive tools and industrial magnetic abrasive tools) were compared and verified.

## 2. Processing Principles

Figure 1 shows a schematic diagram of the wire magnetic abrasive finishing process using ecological magnetic abrasive finishing. In this paper, two different wire magnetic abrasive finishing (WMAF) processes are considered: a WMAF process with ecological magnetic abrasive tools (EMATs) and a WMAF process with industrial magnetic abrasive tools (IMATs). In the EMAT finishing process, a 316L SUS wire workpiece is inserted into the pole gaps of the Nd-Fe-B permanent magnets and ground with the flexible magnetic abrasive brush (FMAB) of the EMAT, which is attached to the sharp edge of the magnetic poles. Both ends of the 316L SUS wire are connected to the drive spools and continuously move with a feed rate of 50 mm/min along with the rotational motion of the drive spools. 

## 3. Experimental Setup and Method

In this study, the surface roughness of 316L SUS wire material was improved by wire magnetic abrasive finishing process. Figure 2 shows a photograph of the wire magnetic abrasive finishing equipment. The wire magnetic abrasive finishing equipment consists of an electrical slider, a motor, two driving motors, two drive spools, several fixing rollers, several controllers, a steel yoke, two sets of Nd–Fe–B permanent magnets, two magnetic poles, 316L SUS wire materials, and the mixture used in the magnetic abrasive tools. Among this equipment, the rotational motion of the finishing part is conducted by the motor, which can be controlled by the controller and enables rotation speeds up to 1200 rpm. To generate the vibration motion of the finishing part, the electrical slider was used, which enables vibration of the finishing part at frequencies up to 10 Hz. During the wire magnetic abrasive finishing process, the 316L SUS wire material was inserted inside the FMAB of ecological magnetic abrasive finishing tools and both ends of the wire were connected to the drive spools. The 316L SUS wire material moved linearly along with the rotational motion of the drive spool at a feed rate of 50 mm/min. Figure 2a shows an enlarged image of the finishing part.

### 3.1. Ecological and Industrial Magnetic Abrasive Tools

Magnetic abrasive tools play an important role in the magnetic abrasive finishing process because they are directly related to the finishing or machining of the surface of the workpiece [10]. In the MAF process, magnetic abrasive tools (i.e., bonded and unbonded types of magnetic abrasives) can be used [11]. The unbonded type of magnetic abrasive does not require a complex preparation process to achieve a high-quality surface finish [12,13]. Therefore, this type of magnetic abrasive was prepared as magnetic abrasive tools. The unbonded type of magnetic abrasive comprises ferrous particles and fine abrasive particles (e.g., aluminum oxide (Al_2_O_3_), silicon carbide (SiC), cubic boron nitride (CBN), carbon nanotubes (CNTs), etc.) [14]. Heng et al. [15] improved the surface roughness of Mg alloy bars using the unbonded type of magnetic abrasives with CNT abrasives, and compared it to the unbonded type of magnetic abrasives without CNT abrasives. Their results showed that when CNT abrasives were used the resulting surface smoothness (Ra) was much greater than when CNT abrasives were not used. Due to their superior mechanical properties, CNT abrasives have been used everywhere in surface finishing methods [16,17]. Besides magnetic abrasives, processing the finishing oil is also known to be a critical input parameter in magnetic abrasive finishing [18,19]. Using magnetic abrasives without processing oil causes micro-scratches to be formed on the finished surface [20]. Therefore, many researchers have studied applications of industrial processing oils to the surface finishing process of materials [21,22,23,24]. However, despite their great potential, side effects on human health have been identified. Tejral et al. [25] reviewed the hazardous effects of CNTs on human health and the environment. They reported that single-wall carbon nanotubes have a potential toxic effect on human health, especially on human peripheral blood lymphocytes. Park et al. [26] reported that industrial oils probably cause more serious danger to human health than vegetable oil. To overcome these problems, new ecological magnetic abrasive tools have been devised to replace industrial magnetic abrasive tools. Ecological magnetic abrasive tools are composed of cuttlefish bone abrasive, olive oil (0.6 mL), electrolytic iron powder Fe#200 (0.9 mg), and diamond abrasive paste Dia: 1-μm (0.45 g). The olive oil is biodegradable, meaning that it is less harmful and renewable. Additionally, it has superior physical properties (i.e., low viscosity, low density) that make it suitable for high-precision finishing of biomaterials. Cuttlefish bone is a natural biomaterial derived from the chamber of the cuttlefish that can be ground into an abrasive or powder. Cuttlefish bone abrasives have many beneficial properties such as being ultra-lightweight while having high stiffness and high permeability [27]. Due to their beneficial properties, they were proposed as raw materials for a range of applications [27]. In particular, they have been used as coating abrasives for polishing amalgam restorations or metal margins. Figure 3a,b shows SEM micro images of cuttlefish bone and carbon nanotube abrasives, respectively.

### 3.2. Material Preparation

In this study, 316L SUS wire (wire dimensions: ø500-μm × Ra. 0.25-μm) was used as the workpiece and its surface accuracy was improved by wire magnetic abrasive finishing using ecological magnetic abrasive tools. 316L SUS wire is a biomaterial that is commonly used for medical applications (e.g., coronary stents, orthodontics, and implantation) due to its superior mechanical properties, physical properties, and anticorrosion properties. Table 1 and Table 2 show the composition and mechanical properties, respectively, of the 316L SUS wire materials used in wire magnetic abrasive finishing.

### 3.3. Experimental Conditions

In this study, our experimental work is separated into two processing stages. Details of the experimental conditions are listed in Table 3. In the first stage, the surface of SUS-316L stent wire was finished by the wire magnetic abrasive finishing process with the ecological magnetic abrasive tools. The mixture used with the ecological magnetic abrasive tools consists of cuttlefish bone abrasives, olive oil (0.6 mL), electrolytic iron powder Fe#200 (0.9 mg), and diamond paste Dia: 0.5-μm (0.45 g). In the second stage the surface of SUS-316L stent wires was finished by the wire magnetic abrasive process with the industrial magnetic abrasive tools. The mixture used with the industrial magnetic abrasive tools consists of CNT abrasives, light oil (0.6 mL), electrolytic iron powder Fe#200 (0.9 mg), and diamond paste Dia: 0.5-μm (0.45 g). To improve the processing efficiency, two stages of wire magnetic abrasive finishing process were performed with different magnetic field vibration frequencies (5, 10 Hz) and different magnetic field rotational speeds (750, 950, 1200 rpm). To quantify changes in surface improvement, the surface finish of each workpiece was measured at each cycle using a surface roughness tester. Atomic force microscopy (AFM) and SEM micro images also were used to compare the finishing capacity of wire magnetic abrasive finishing with two different magnetic abrasive tools.

## 4. Results and Discussions

### 4.1. Finishing Characteristics of Ecological Magnetic Abrasive Tools

#### 4.1.1. Effect of Rotational Speed

To understand the finishing characteristics of ecological magnetic abrasive tools, SUS-316L stent wires were improved by wire magnetic abrasive finishing process, which was performed with the ecological magnetic abrasive tools mixture under different rotational speeds (750, 950, and 1200 rpm). Figure 4 shows the correlation between surface roughness and number of cycles for different rotation speeds of the magnetic field. The results observed in Figure 4 show that the original surface roughness of the workpieces was dramatically improved by all rotation speeds of the magnetic field. Among the three conditions, the highest rotation speed of 1200 rpm is observed to be the best parameter in this experiment. Conversely, the least surface improvement was observed for the 750 rpm rotation speed. By increasing the rotation speed parameter, the surface roughness per number of cycles decreases, resulting in a better surface finish. The result can be explained due to the increasing of rotational speed. When the rotational speed of magnetic field increases to 1200 rpm, the magnetic abrasive tools have much time for cutting the unevenness of wire workpiece surface, thus more improvement in surface roughness value was observed. It can be seen from Figure 4 that for all rotation speeds, the surface roughness decreased rapidly during the first finishing cycle (No. 1), and then continued to decrease until the final cycle (No. 4). Thus, it can be confirmed that the surface roughness of the workpiece decreased with increasing numbers of finishing cycles for the given experimental conditions. When the highest rotation speed (1200 rpm) was used, the initial surface roughness was decreased from 0.23 μm to 0.06 μm.

#### 4.1.2. Effect of Magnetic Field Vibration Frequency

Figure 5 shows the correlation between surface roughness and number of finishing cycles for different magnetic field vibration frequencies. To find the optimal vibration frequency, the workpiece was improved at 5 Hz and 10 Hz at 1200 rpm. The results showed that the surface roughness of SUS-316L stent wires decreased with both vibration frequencies (5 Hz and 10 Hz). However, the decrease in surface roughness is more significant when a vibration frequency of 10 Hz was used. When the vibration frequency increases to 10 Hz, the repeated number of cutting between the magnetic abrasive tools and workpiece surface increases, and thus more improvement in surface roughness value was observed. The initial surface roughness of 0.23 μm was decreased to 0.06 μm at the fourth finishing cycle when 10 Hz was used. In contrast, the initial surface roughness of 0.22 μm was decreased to 0.08 μm at the fourth finishing cycle when 5 Hz was used.

### 4.2. Finishing Characteristics of Industrial Magnetic Abrasive Tools

#### 4.2.1. Effect of Rotational Speed

To understand the finishing characteristics of industrial magnetic abrasive tools, SUS-316L stent wires were improved by the wire magnetic abrasive finishing process, which was performed with the industrial magnetic abrasive tools mixture under different rotational speeds (750, 950, and 1200 rpm). Figure 6 shows the correlation between surface roughness and cycle number for different rotation speeds when industrial magnetic abrasive tools were used. Similar to the observations made under the same conditions for ecological magnetic abrasive tools, the original surface roughness of 316L stent wires was dramatically improved by all tested magnetic field rotation speeds. However, the decrease in surface roughness was more significant when the highest rotation speed (1200 rpm) was used. Conversely, when a rotation speed of 750 rpm was used, the surface improvement was less than under the other two conditions. The result can be explained due to the increasing of rotational speed. When the rotational speed of magnetic field increases to 1200 rpm, the magnetic abrasive tools have much time for cutting the unevenness of wire workpiece surface. The best improvement in surface roughness is observed when 1200 rpm was used, followed by 950 rpm, and 750 rpm. When the highest rotation speed of 1200 rpm was used, the initial surface roughness decreased from 0.23 μm to 0.05 μm at the third finishing cycle.

#### 4.2.2. Effect of the Magnetic Field Vibration Frequency

Figure 7 shows the correlation between surface roughness and finishing cycle number for different magnetic field vibration frequencies. To determine the optimal vibration frequency, the workpiece was improved at 5 Hz and 10 Hz at 1200 rpm with the industrial magnetic abrasive tools. As with ecological magnetic abrasive tools, the greatest improvement in surface roughness is observed when a vibration frequency of 10 Hz was used. When a magnetic field vibration frequency of 10 Hz was used, the surface roughness decreased from 0.23 μm initially to 0.05 μm at the third finishing cycle. Conversely, at 5 Hz, the initial surface roughness of 0.23 μm decreased to 0.07 μm at the third finishing cycle.

### 4.3. Comparison

Figure 8 shows the correlation between surface roughness and number of finishing cycles for the different magnetic abrasive tools. The surface roughness of 316L stent wires decreased with both conditions. However, the surface improvement by industrial magnetic abrasive tools is found to be better than that of ecological magnetic abrasive tools. This could be explained by the excellent mechanical properties of CNT abrasives and light oil. CNT abrasives have a higher tensile strength and a higher elastic modulus than other abrasive particles (e.g., Al_2_O_3_, SiC, Boron, and CFB abrasive) [28,29]. Additionally, light oil was used as the grinding oil in this experiment. It has a lower viscosity and density than olive oil. Nurul et al. [30] concluded that low viscosity grinding oil could reduce the heat generated in the finishing process better than high viscosity grinding oil. Table 4 shows the mechanical properties of grinding oils used in the wire magnetic abrasive finishing process. When the industrial magnetic abrasive tools were used, the surface roughness (Ra) of 316L stent wires decreased from 0.23 μm to 0.05 μm; when ecological magnetic abrasive tools were used, the surface roughness decreased from 0.23 μm to 0.06 μm.

Figure 9 shows SEM and AFM surface images of 316L stent wire before finishing by wire magnetic abrasive finishing process. As can be seen from Figure 9, unevenness can be seen throughout the original surface of the 316L stent wire and the original surface roughness Ra before finishing was 0.23 μm. Figure 10 shows SEM and AFM surface images of 316L stent wire after finishing by ecological magnetic abrasive tools. Based on Figure 10, it could be concluded that the ecological magnetic abrasive tools have the ability to achieve a high surface quality of 316L SUS wire workpiece. The final surface roughness achieved by ecological magnetic abrasive tools was 0.06 μm. Figure 11 shows SEM and AFM surface images of 316L stent wire after finishing by the industrial magnetic abrasive tools. As can be seen from Figure 11, the finished surface of the workpiece is much smoother than the initial surface in Figure 9 and its surface topography is similar to that of the finished surface in Figure 11. Figure 12 shows the chemical elements of 316L SUS wire before finishing with the ecological magnetic abrasive tools. The energy-dispersive X-ray (EDX) analysis determined a surface composition of 41.14% C, 1.45% Si, 0.12% P, 3.85% Mo, 33.37% Cr, 2.60% Mn, and 17.47% Ni for the 316L SUS wire workpiece. Figure 13 shows the chemical elements of 316L SUS wire after finishing by the ecological magnetic abrasive tools; the energy-dispersive X-ray (EDX) analysis determined a surface composition of 38.73% C, 1.52% Si, 0.19% P, 4.15% Mo, 34.80% Cr, 2.56% Mn, and 18.05% Ni for the 316L SUS wire workpiece. The results of the EDX analysis indicate that the components of the ecological magnetic abrasive tools were not detected at the surface of the 316L SUS wire workpiece.

## 5. Conclusions

In this paper, we develop a new ecological magnetic abrasive tool for finishing wire material using a wire magnetic abrasive finishing process.
When the ecological magnetic abrasive tools were used, the initial surface roughness of 316L SUS wires decreased from 0.23 μm to 0.06 μm after four finishing cycles with a 1200 rpm magnetic field rotation speed and 10 Hz vibration frequency.In contrast, the industrial magnetic abrasive tools decreased the surface roughness from 0.23 μm to 0.05 μm after four finishing cycles using the same conditions. The least surface roughness was measured after treatment with the industrial magnetic abrasive tools. This can be attributed to the excellent mechanical properties of industrial magnetic abrasive tools when compared to the ecological magnetic abrasive tools.Based on the result of EDX analysis, it was confirmed that the components of ecological magnetic abrasive tools were not detected at the surface of 316L SUS wire workpiece.Based on the results, it can be concluded that replacing industrial magnetic abrasive tools with ecological magnetic abrasive tools is achievable.

## Figures and Tables

**Figure 1 materials-12-00714-f001:**
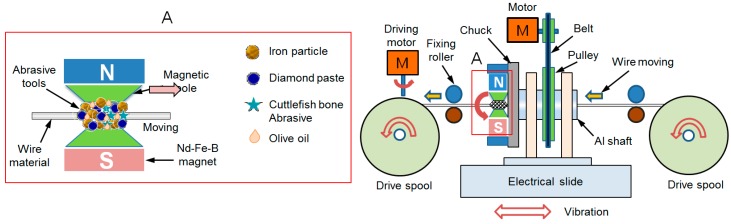
Schematic diagram of the wire magnetic abrasive finishing process using ecological magnetic abrasive finishing.

**Figure 2 materials-12-00714-f002:**
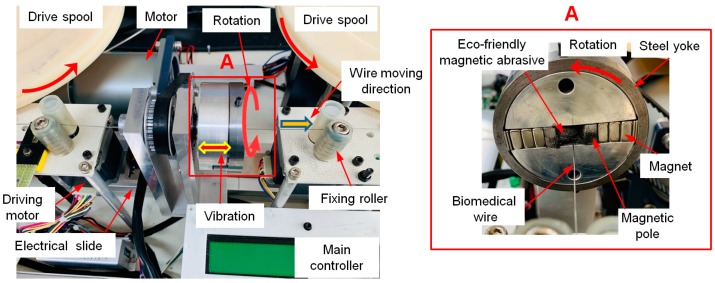
Photograph of wire magnetic abrasive equipment. (**A**) Enlarged image of finishing part.

**Figure 3 materials-12-00714-f003:**
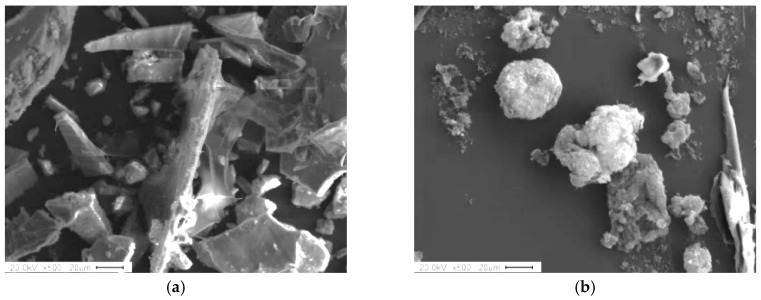
SEM micro images of abrasive tools used in wire magnetic abrasive finishing process. (**a**) SEM micro image of cuttlefish bone abrasives; (**b**) SEM micro image of CNT abrasives.

**Figure 4 materials-12-00714-f004:**
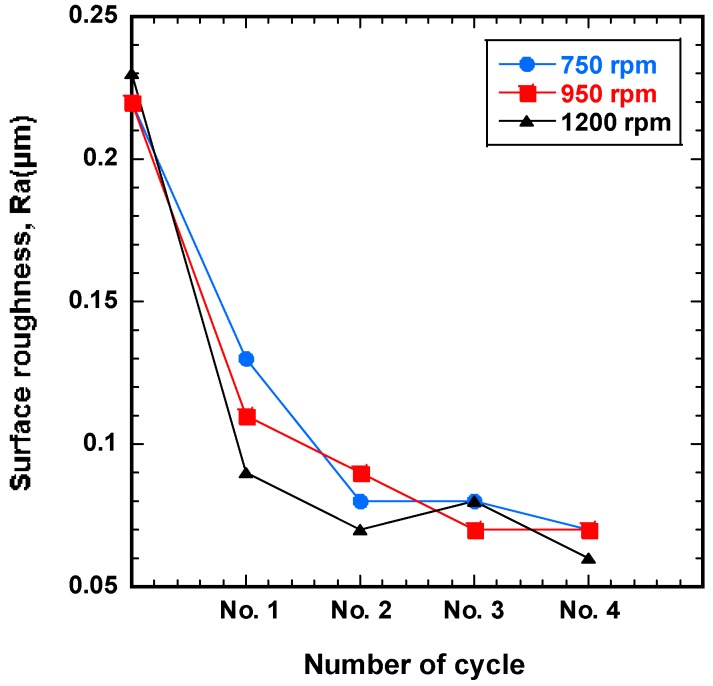
Surface roughness (Ra) as a function of the number of finishing cycles for different magnetic field rotation speeds with eco-friendly magnetic abrasive tools.

**Figure 5 materials-12-00714-f005:**
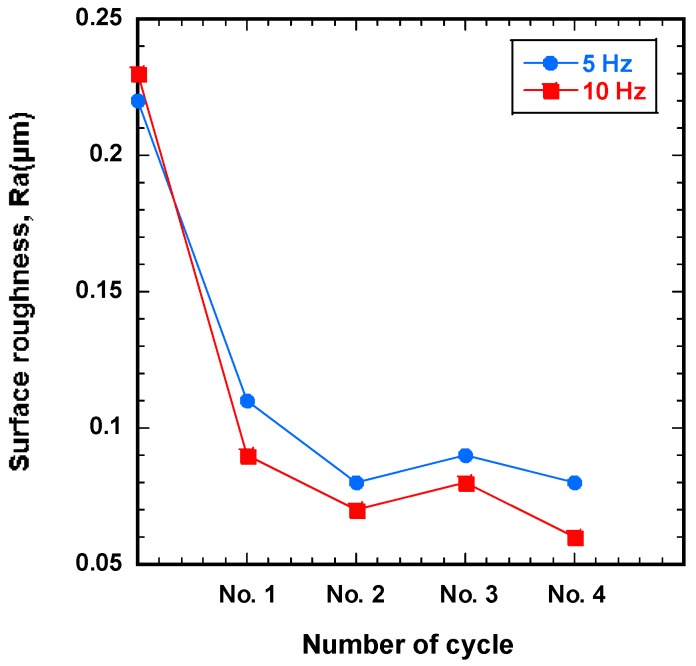
Surface roughness (Ra) as a function of the number of finishing cycles for different magnetic field vibration frequencies with eco-friendly magnetic abrasive tools.

**Figure 6 materials-12-00714-f006:**
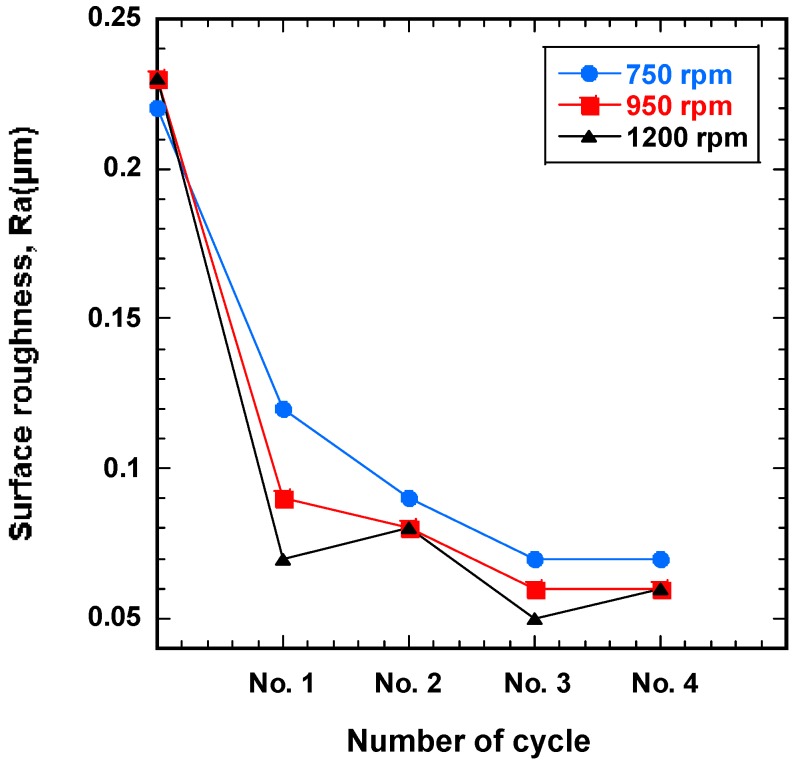
Surface roughness (Ra) as a function of the number of finishing cycle for different magnetic field rotation speeds with industrial magnetic abrasive tools.

**Figure 7 materials-12-00714-f007:**
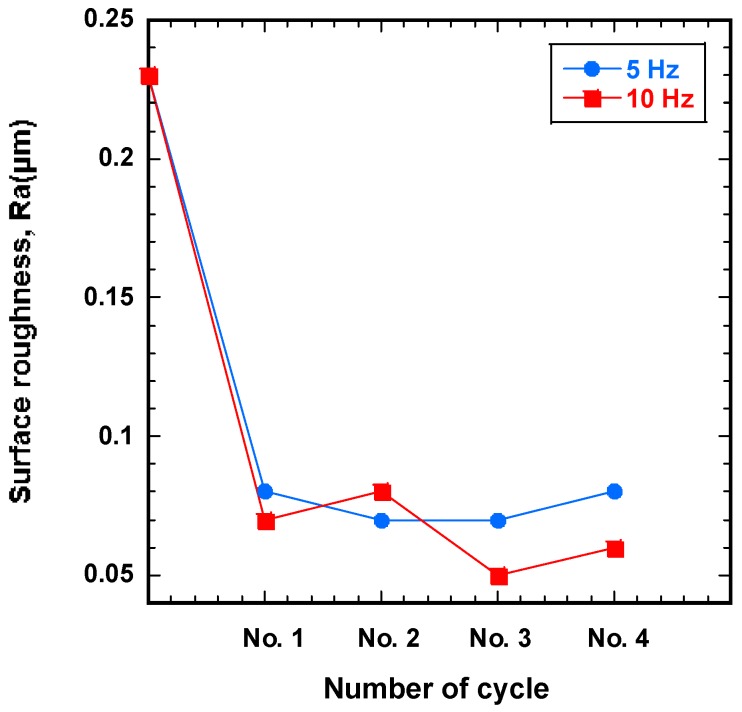
Surface roughness (Ra) as a function of the number of finishing cycle for different magnetic field vibration frequencies with industrial magnetic abrasive tools.

**Figure 8 materials-12-00714-f008:**
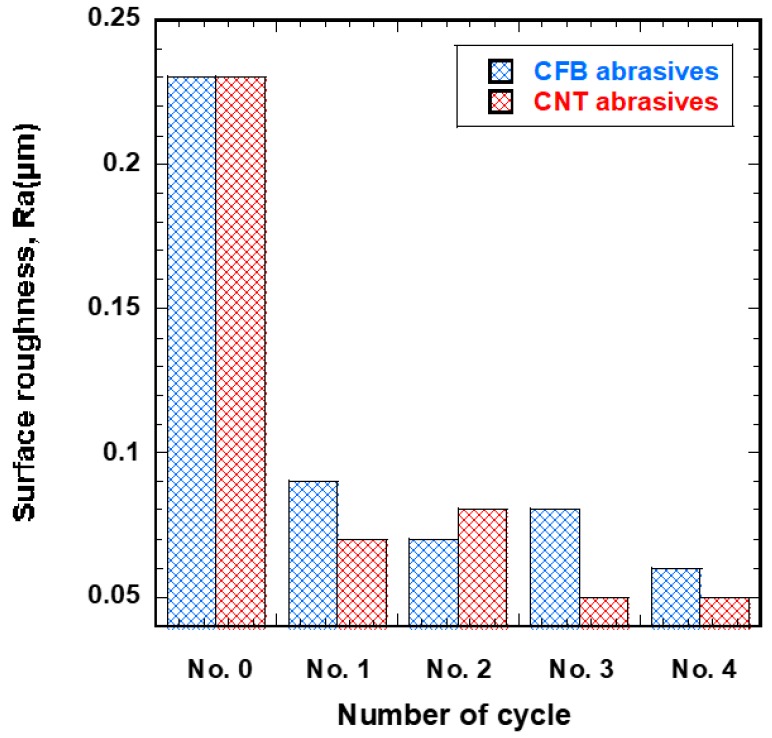
Correlation between surface roughness, Ra, and number of finishing cycles for different magnetic abrasive tools.

**Figure 9 materials-12-00714-f009:**
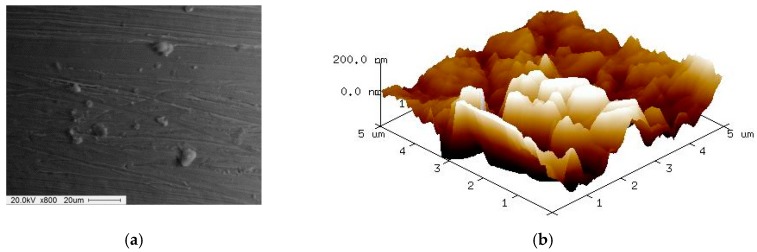
SEM and AFM surface images of 316L stent wires (Ra: 0.23 μm). (**a**) SEM surface image before finishing; (**b**) AFM surface image before finishing.

**Figure 10 materials-12-00714-f010:**
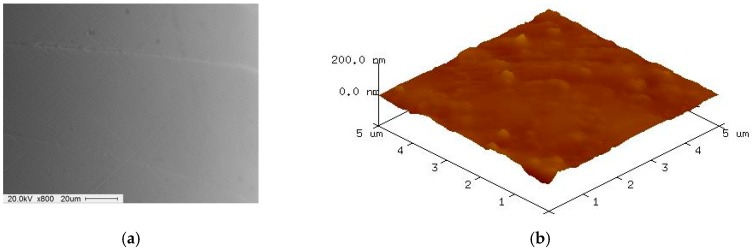
SEM and AFM surface images of 316L stent wire after finishing by ecological magnetic abrasive tools (Ra: 0.06 μm, 1200 rpm, 10 Hz). (**a**) SEM surface image after finishing; (**b**) AFM surface image after finishing.

**Figure 11 materials-12-00714-f011:**
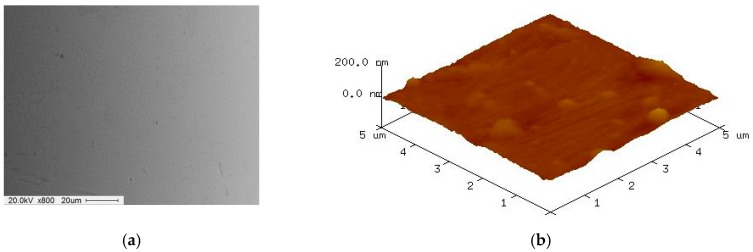
SEM and AFM surface images of 316L SUS wire after finishing by industrial magnetic abrasive tools (Ra: 0.05 μm, 1200 rpm, 10 Hz). (**a**) SEM surface image after finishing; (**b**) AFM surface image after finishing.

**Figure 12 materials-12-00714-f012:**
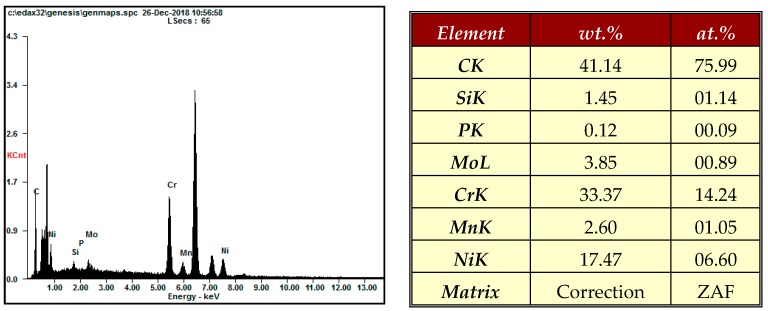
Chemical elements of 316L SUS wire before finishing.

**Figure 13 materials-12-00714-f013:**
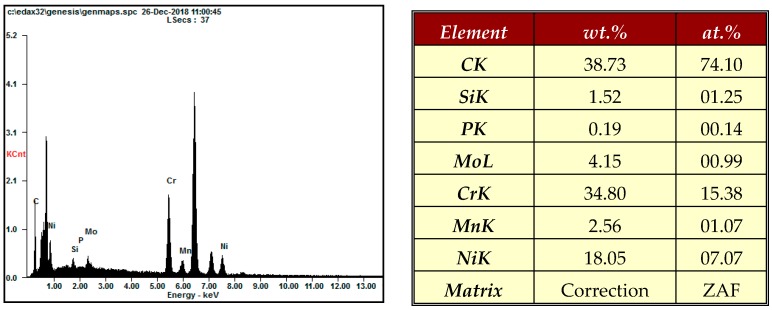
Chemical elements of 316L SUS wire after finishing with ecological magnetic abrasive tools (Ra: 0.06 μm, 1200 rpm, 10 Hz).

**Table 1 materials-12-00714-t001:** Chemical elements 316L SUS wire.

Component	C	Mn	Si	Cr	Ni	P	Mo
SUS-316L (Max%)	0.03	2	1	16–18.5	10–14	0.045	3

**Table 2 materials-12-00714-t002:** Mechanical properties of AISI 316L SUS wire.

Density	Modulus of Elasticity	Specific Heat	Thermal Conductivity	Electrical Resistivity	Melting Range
8.07 Mg/m^3^	310 MPa	450 J/kg-K (0–100 °C)	14.6 W/m-K	74 μΩ-cm at 20 °C	1390–1440 °C

**Table 3 materials-12-00714-t003:** Experimental conditions for both finishing processes.

Constant Parameters	EMAT	IMAT
Workpiece	SUS-316L wire (wire dimensions: ø500-μm × Ra. 0.25-μm)	SUS-316L wire (wire dimensions: ø500-μm × Ra. 0.23-μm)
Magnetic particles	Electrolytic iron powder: Dia: 200 μm (0.90 g) Diamond paste: Dia: 0.5 μm (0.45 g)	Electrolytic iron powder: Dia: 200 μm (0.90 g) Diamond paste: Dia: 0.5 μm (0.45 g)
Abrasives	Cuttlefish Bone Powder (3000 mm^3^)	Carbon nanotube (3000 mm^3^)
Grinding oil	vegetable oil (Olive oil-Fontana) 0.6 mL	Industrial oil (Light oil-WD40) 0.6 mL
Vibration frequency	5, 10 Hz	
Magnetic pole shapes	Sharp edge	
Rotational speed	750, 950, 1200 rpm	
Magnetic flux density at finishing point	495 mT	
Number of cycles	4	

**Table 4 materials-12-00714-t004:** Mechanical properties of grinding oils used in ire magnetic abrasive finishing.

Grinding Oil	Viscosity (Pa-s)	Density (Kg/cm^3^)	Surface Tension (N/m)
Olive oil	0.0437	957	0.01
Light oil	0.05	800–820	0.031

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
