# Peer review of "Development of a New Ecological Magnetic Abrasive Tool for Finishing Bio-Wire Material"

_materials, 2019, doi:10.3390/ma12050714_

Reviewer 1 Report

The article presents way of substituting abrasive materials with biodegradable materials in the magnetic abrasive finishing process. These tests are important and interesting due to problems occurring during  this type of machining. This article develops knowledge enabling effective processing with using biodegradable materials.

Remarks on the article:

Biodegradable materials have long been used in abrasive machining. For example, crumbled shells of nuts are commonly used for vibratory finishing of soft metals and their alloys. In the introduction may be mentioned to this type of finishing processes and its results.

Figure 3 does not provide much information and may be replaced with a table that describes in detail the components used, along with theirs percentage share in the working material applied.

Figure 5 does not provide any information and should be removed as the appearance of the 0.5 mm wire is commonly known.

SI units should be used in the work as a basic value, notwithstanding may be supported by other units, saved as an additional value near basic value.

In table 2 there are no units for density. Table 3 should be supplemented by the magnitude of the magnetic field at the machining point.

In the work are unintelligible statements, for example: in lines 178-181 is talked about the centrifugal force which pushes abrasive particles in direction to the surface. This is incomprehensible because the grains should be pressed against the external surface while the centrifugal force works in the other direction. (The principle described in the article works in the case where is machined an internal surface, e.g. a hole). A similar statement is in lines 197-198 where an increasing coefficient of friction is associated with a better (smoother) surface.

In Figures 6-10, the standard deviation of measurements should be marked.

In Table 4, the oil should be supplemented with the commercial name because the oil producers uses additives that may affect the test results.

In Figures 11-13, in parts “a”, the magnification should be larger so that the wire surface structure can be noticed. In turn, in parts “b”, the larger magnification should refer to the z axis or to used more diverse colours of the contour layers.

In line 281 is recommended to avoid grammatical structures in the first person.

The influence of vibration and rotational motion is slightly unreadable at work. It would be necessary to specify precisely what effect only vibrations and only rotational motion have on the results of the process. Tabulation of technological data and machining results is recommended. In the case of vibrations, the amplitude and plane of vibrations should be given.

Author Response

Dear Editor and Reviewers,

We would like to thank the editor and reviewers for careful and thorough reading of this manuscript and for the thoughtful comments and constructive suggestions, which help to improve the quality of this manuscript.

Manuscript: Development of a new ecological magnetic abrasive tool for finishing bio-wire material

Reviewer# 1

1- Comment: Figure 3 does not provide much information and may be replaced with a table that describes in detail the components used, along with theirs percentage share in the working material applied.

Response: Thank so much for your kind comment. As your comment I have removed Figure 3 from my manuscript. The detail component used of ecological magnetic abrasive tools and industrial magnetic abrasive tools are already listed in Table 3, and we used weight and volume as their amount not percentage. As highlighted in yellow color in Table 3.

2- Comment: Figure 5 does not provide any information and should be removed as the appearance of the 0.5 mm wire is commonly known.

Response: As your comment, I have already removed Figure 5 from my manuscript.

3- Comment: SI units should be used in the work as a basic value, notwithstanding may be supported by other units, saved as an additional value near basic value.

Response: Thank you for your kind comment. As, your comment, all the units were already revised to SI uinit. You can find in Table 2, and Table 4 highlighted in yellow color.

4- Comment: In table 2 there are no units for density. Table 3 should be supplemented by the magnitude of the magnetic field at the machining point

Response: Thank you for your comment.

1.     I already added the units for density. It can be found in Table 2.

2.     The Magnetic flux density at finishing point is already supplied in Table 3.

5- Comment: In the work are unintelligible statements, for example: in lines 178-181 is talked about the centrifugal force which pushes abrasive particles in direction to the surface. This is incomprehensible because the grains should be pressed against the external surface while the centrifugal force works in the other direction. (The principle described in the article works in the case where is machined an internal surface, e.g. a hole). A similar statement is in lines 197-198 where an increasing coefficient of friction is associated with a better (smoother) surface

Response: You are right. Thank you so much for your comment. I already revised the statement in lines 178-181 and lines (197-198)

After revised:

The result can be explained due to the increasing of rotational speed. When the rotational speed of magnetic field increased to 1200 rpm, the magnetic abrasive tools have much time to finish the unevenness of wire workpiece surface, thus more improvement in surface roughness value was observed.

When the vibration frequency increases to 10 Hz, the repeated number of friction between the magnetic abrasive tools and workpiece surface increased, and thus more improvement in surface roughness value was observed.

6- Comment: In Figures 6-10, the standard deviation of measurements should be marked.

Response: Thank you for your comment. I cannot mark the standard deviation of measurements to mark on Figures 6-10, because my graph program does not have function to do that. 

7- Comment: In Table 4, the oil should be supplemented with the commercial name because the oil producers uses additives that may affect the test results.

Response: As your comment we have already added the commercial name of both oils i.e., vegetable oil (Olive oil-Fontana), and industrial oil (Light oil-WD40)

8- Comment: In Figures 11-13, in parts “a”, the magnification should be larger so that the wire surface structure can be noticed. In turn, in parts “b”, the larger magnification should refer to the z axis or to used more diverse colours of the contour layers

Response: Thank you for your comment, as your comment, I have already changed my Figure 11-13 in part a, and b.

9- Comment: In line 281 is recommended to avoid grammatical structures in the first person.

Response:  As your comment, I have revised from “chemical composition” to “chemical elements”.

10- Comment: The influence of vibration and rotational motion is slightly unreadable at work. It would be necessary to specify precisely what effect only vibrations and only rotational motion have on the results of the process. Tabulation of technological data and machining results is recommended. In the case of vibrations, the amplitude and plane of vibrations should be given.

Response: I have already revised.

In the case of rotational speed:

The result can be explained due to the increasing of rotational speed. When the rotational speed of magnetic field increased to 1200 rpm, the magnetic abrasive tools have much time to finish the unevenness of wire workpiece surface, thus more improvement in surface roughness value was observed.

In the case of vibration frequency:

When the vibration frequency increases to 10 Hz, the repeated number of friction between the magnetic abrasive tools and workpiece surface increased, and thus more improvement in surface roughness value was observed.

Reviewer 2 Report

Please do a little effort following all advises in attached file.

Author Response

Dear Editor and Reviewers,

We would like to thank the editor and reviewers for careful and thorough reading of this manuscript and for the thoughtful comments and constructive suggestions, which help to improve the quality of this manuscript.

Manuscript: Development of a new ecological magnetic abrasive tool for finishing bio-wire material

Reviewer #2

1- Comment: Conclusions: too, brief, give them as bullets, one per each highlight.

Response: Thank so much for your kind comment. As your comment, I have already revised “my Conclusions Section”, by using the order number from 1 to 4.

2- Comment: State of the art: MDPi is looking for contributions in this type of processes, recently: J. Manuf. Mater. Process. 2018, 2(4), 82; doi:10.3390/jmmp2040082 , for instance, gave several new ideas about abrasives. The experiments were performed in part on the Machining Science and Technology 16 (2), 173-188.

Response: As your comment, I have already used these papers as my references.

4. Olvera, D.; Rodríguez, C.A.; Elías-Zúñiga, A.; Urbikain, G.; Lacalle, L. Helical milling for hole making in nickel-based alloy with ball-end mill; 2009; Vol. 37, pp. 253-260.

14. Rodríguez, A.; Fernández Valdivielso, A.; Lacalle, L.; Sastoque Pinilla, L. Flexible Abrasive Tools for the Deburring and Finishing of Holes in Superalloys; 2018; Vol. 2, pp. 82.

3- Comment: Force feedback control is the key aspect to be considered in polishing. Please include some discussion, making stress in your approach. The paper makes a rapid view on ecological and life cycle. However, some extended ideas can be.

Response: Thank you for your kind comment. In our approach study will be included with the force, and stress.

4- Comment: Abrasive works on additive parts can be also key in the future. Optics and Lasers in Engineering 56, 113-120, is an initial point. The use of techniques in this line can be astonishing!!

Response: Thank you for your comment. In our approach study will be included with this paper.

5- Comment: Journal of cleaner production is missed…it is the key journal in the matter. For instance works by Pereira about life cycle analsysys, or MQL or coolants or CO2 are missed.

Response: As your comment, I have already used these papers as my references.

6. Rodriguez, A.; Lacalle, L.; Calleja, A.; Fernández Valdivielso, A.; Lamikiz, A. Maximal reduction of steps for iron casting one-of-a-kind parts; 2012; Vol. 24, pp. 48-55.

Round  2

Reviewer 1 Report

Figure 9b has cut out some data. 

The work presents two types of processing at the same time. Vibratory finishing and machining by turning the abrasive around the wire. In point 10 I asked what the effects of machining will be when only one of its types is applied, eg vibrations will be switched off. 

I suggest replacing in line 193 word " friction" with the word "cutting" or "plowing"

Author Response

Dear Editor and Reviewers,

We would like to thank the editor and reviewers for careful and thorough reading of this manuscript and for the thoughtful comments and constructive suggestions, which help to improve the quality of this manuscript.

Manuscript: Development of a new ecological magnetic abrasive tool for finishing bio-wire material

1- Comment:

Figure 9b has cut out some data.

The work presents two types of processing at the same time. Vibratory finishing and machining by turning the abrasive around the wire. In point 10 I asked what the effects of machining will be when only one of its types is applied, eg vibrations will be switched off.

I suggest replacing in line 193 word " friction" with the word "cutting" or "plowing"

Response:

1.      Thank you for your kind comment. As your comment I have added one sentence to Figure 9, as highlighted in yellow color.

2.      It is a good question. In magnetic abrasive finishing or in surface machining process, both a relational motion of abrasives and a vibration of workpiece (i.e., frequency and amplitude) are important during the processing. Because without vibration and with only rotational motion, circumferential grooves will form on the surface after processing.

3.      As the reviewer comment. I have revised from "friction" to "cutting" (Page: 6, Line 193)

Reviewer 2 Report

Good revision

Author Response

We would like to thank the editor and reviewers for careful and thorough reading of this manuscript and for the thoughtful comments and constructive suggestions, which help to improve the quality of this manuscript.

Best regards